# Is the Introduction into a New Environment Stressful for Young Bulls?

**DOI:** 10.3390/vetsci10090545

**Published:** 2023-08-30

**Authors:** Martina Moriconi, Giulia Pagliasso, Francesca Fusi, Nicoletta Vitale, Lisa Guardone, Mario Vevey, Alessandro Dondo, Elisabetta Razzuoli, Stefania Bergagna

**Affiliations:** 1Istituto Zooprofilattico Sperimentale del Piemonte, Liguria e Valle d’Aosta, Via Bologna148, 10154 Torino, Italylisa.guardone@izsto.it (L.G.);; 2Azienda Sanitaria Locale di Ciriè, Chivasso, e Ivrea, Via Cavour 29, 10073 Ciriè, Italy; 3Italian National Reference Centre for Animal Welfare (CReNBA), Istituto Zooprofilattico Sperimentale della Lombardia e dell’Emilia Romagna “Bruno Ubertini” (IZSLER), Via A. Bianchi 9, 25124 Brescia, Italy; 4Associazione Nazionale Bovini di Razza Valdostana, Fraz. Favret, 5, 11020 Gressan, Italy

**Keywords:** new environment, stress, immune response

## Abstract

**Simple Summary:**

A stressor is an internal or external stimulus or threat (physical, psychological, or chemical) that disrupts homeostasis and activates a response to cope with a threat and return to or maintain homeostasis. Hematological investigations and the evaluation of innate immune response and inflammation are methods that can be used to measure an animal’s response to stressful stimuli. The aim of this study is to evaluate hemato-chemical and immunological parameters for 45 young bulls in order to determine whether the introduction into a new environment, after separation from their dams, transportation from the farm of origin to a genetic center, and regrouping with other animals can alter them. The results obtained show an acute phase response and activation of innate immune responses, suggesting a mild response to adaptation stress by calves after the introduction into a new environment.

**Abstract:**

Several events in an animal’s life are considered stressful. Among them, the most studied and significant are transportation, weaning, and adaptation to climate change. Moreover, other events, such as the separation from the dam, moving from the original farm to another, management practices, such as regrouping with other animals, and new hierarchical conditions, represent routine conditions in the bovine’s life, which can influence the animal’s homeostasis. The purpose of this study is to evaluate the changes in blood parameters of 45 calves introduced into a new environment from their original farms. Blood samples were collected upon arrival at a genetic center (T1), 7 (T2), 30 (T3), and 120 (T4) days after arrival. Blood count, protein electrophoresis, clinical chemistry, and innate immunity parameters were performed on the samples. Significant alterations in some clinical chemistry parameters were related to liver function in the serum protein and the values of IL-6 and TNF-α; the main cytokines mediating the stress response emerged from the results. The evidence indicates the mild response to adaptation stress by calves raised in close association with people after their introduction into a new environment.

## 1. Introduction

Any external or internal stimuli or threat (e.g., chemical, physical, or psychological) that disrupt body homeostasis can be defined as a “stressor” [1,2,3]. In response to such an alteration, a stress response is activated to help the body return to its physiological condition [4]. The negative impact of stress on animal health, well-being, and productivity is widely accepted among livestock breeders and scientists [5]. 

Behavioral and physiological changes are needed to respond to stress; this is characterized by endocrine changes in the hypothalamus–pituitary–adrenal axis (HPA), with a transient increase in the hormone cortisol, which can modulate a variety of biological effects in the body, including immune functions. Acute stress can trigger the immune system to prepare for a potential pathogen invasion and subsequent infection [5]. In the case of chronic stress, cortisol also prevents the excessive stimulation of the immune system by the suppression of T- and B-lymphocyte blastogenesis, as well as natural killer cell activity, cytokines (interleukin-2 (IL-2), interleukin-6 (IL-6), tumor necrosis factor-α (TNF-α), and interferon-γ (IFN-γ)) [5,6,7] production, and reduces inflammation, which can have both a beneficial or detrimental effect on the animal [5].

Reproduction, growth, and immune functions can be adversely affected by an increased secretion of hormones related to stress. This condition can occur due to a response determined by non-infective stressors associated with livestock management procedures, such as transportation, regrouping, weaning, and castration [1,8]. In cattle research, several studies have investigated the stress-related immune function and hormones focusing on animal husbandry [9,10,11] and environmental parameters, such as temperature, temperature–humidity index, and relative humidity [12,13,14]. 

The health and welfare of farm animals are highly influenced by stress factors, such as transportation, which includes both physical and psychological stimuli that can lead to possible negative repercussions [15]. The transport of cattle can result in immune suppression, which can lead to increased susceptibility to diseases and can result in an increased spread of pathogens as well as high mortality during the first few days of transport. Investigations on the biological response to transportation showed stress-related alterations in hematological and inflammatory responses in cattle after transport [9,15,16,17,18,19,20].

Weaning is an essential step in animal breeding; however, it also represents a multifactorial stressor due to dietary change, maternal separation, social disruption, and movement to a new environment [21]. The weaning of beef calves reared in large systems usually occurs around six months old [22], while dairy farm calves are generally separated from their mothers immediately or a few hours after birth [23]. Abrupt weaning causes acute and persistent stress to the calf, including multiple stressors, such as the loss of the mother, as well as changes in social and physical environments [24,25]. Studies conducted to observe the differences in calves’ immune responses have shown that housing altered their neutrophil and lymphocyte immunophenotypes, along with the acute phase response [22]. A stronger stress response was observed in calves weaned alone in comparison with those housed with their dams [22,26]. Modifications in calf immunity due to weaning are relevant as they are thought to be associated with the increased incidence and severity of respiratory diseases [27].

In recent years, the effects of climate change on the animals’ ability to adapt to new climatic conditions has attracted considerable attention. The animals’ immune systems and productivity are affected by heat stress (HS). Environmental temperature and humidity are important factors affecting HS severity [28] and must be monitored and possibly controlled when reaching extreme values to prevent animals’ metabolic stress.

Few studies, however, have considered novel environment, new commingling, feeding, and hierarchy conditions as stressful events [6]. Integrated welfare-friendly practices should cover the whole life of farm animals, including all the practices that superficially can be considered less relevant than others, such as a herd’s new conditions after its introduction into a new environment. Nevertheless, the innate immune system is able to respond to both infectious stressors (microbial agents) as well as non-infectious stressors (high/low temperatures, psychological stress, endocrine disrupters, oxidative stress, tissue damage, hypoxia, obesity, etc.) [6].

In addition to the importance of a single stressor, the cumulative effects of multiple stressors should be considered, as some of these may be multiplicative rather than additive. The aim of this study is to evaluate the trend of hematological, hemato-chemical, and immunological markers of young bulls after separation from their dams, moving from their original farms and being introduced into a new environment, as all of these factors, alone or in combination, are considered important stressors in animals. 

## 2. Materials and Methods

The Ethical Committee on Animal Use of the Istituto Zooprofilattico Sperimentale of Piemonte, Liguria and Valle d’Aosta approved the current study with protocol n° 3272. 

### 2.1. Study Design

The study involved 45 bulls selected from 40 farms, all located in the Valle d’Aosta region. They belonged to two autochthonous Valdostana cattle breeds, the Aosta Black Pied and Aosta Red Pied (Valdostana Pezzata Nera and Valdostana Pezzata Rossa, respectively), farmed for both their meat and milk in a semi-intensive system characterized by mountain pastures in the summer season and recovery in tie-stall farms during winter [29]. The national association of Valdostana breeders, called “Associazione Nazionale Allevatori Bovini di Razza Valdostana” (A.N.A.Bo.Ra.Va.) in Italian, is based in Gressan in the province of Aosta. The association promotes various activities, including the genetic improvement of the Valle d’Aosta cattle breed, evaluations of the reproducers, organization of national and foreign exhibitions, management of the genetic center for performance tests, and management of the bull center, where the semen of bulls used for artificial insemination is produced [30]. At the age of 45–60 days old, the young bulls are sent to the A.N.A.Bo.Ra.V.A. control center. Here, after weaning is performed on animals that are 130 and 150 days old, the performance tests are begun. At around 11 months old, if they pass the tests, they are sent to the artificial insemination center to start breeding activity.

The animals selected for transportation to the genetic center were obtained from small, fixed-stall family farms. They were valuable animals, registered in the breed studbook and accustomed to the daily care and attention of breeders who were particularly sensitive to animal health and welfare. Since birth, the calves remained under their dams for at least 21 days, after which they were removed and raised in single stalls and subjected to the care and attention of the breeders focusing on nutrition, environmental conditions, comfort, and handling. The bulls were transported by a commercial truck equipped with dividers to reduce space and to prevent falls and crushing, with a maximum capacity of 12 calves; an adequate layer of chopped straw litter was used to cover the truck floor and both natural and mechanical ventilation methods was adopted. Each animal was transported individually from the original farm to the control center, with the exception of 10 animals, which were transported in groups of two as they came from the same farm. Upon the arrival at the genetic center, the animals’ weight was 84.2 ± 12.13 kg and they were 60.7 ± 8.5 days old. All bulls received antiparasitic treatment at the same time (Virbamec, Virbac, Milan), while vaccination (Rispoval Intranasal Rs+PI3, Zoetis, Rome) was performed within 10–20 days of their arrival at the center.

Blood was collected from the animals when they arrived at the control station (T1) to define the baseline values, 7 (T2), 30 (T3), and 120 (T4) days from arrival. Tubes containing anticoagulant (K3EDTA) were used for jugular sampling whole blood samples, which were immediately processed for their hematological count as soon as they arrived at the laboratory. Serum samples were obtained using tubes not containing anticoagulant, which were then centrifuged (3500 rpm, 15 min, room temperature). Then, the serum fraction was transferred into a tube and 2 mL aliquots were stored at −80 °C. The biochemical profile, protein electrophoresis test, and evaluation of innate immunity parameters were performed on the serum samples. 

### 2.2. Blood Analysis

Whole blood samples were processed within 24 h of collection on a Malet Schloesing^®^-MSA instrument (Melet-Schloesing-MS4, Osny, France), as described in the manufacturer’s instructions. The investigated parameters were leukocytes (WBCs) with a leukocyte formula (monocytes (MONs), lymphocytes (LINFs), neutrophils (NEUs), eosinophils (EOs), and basophils (BAs)) and platelets (PLTs), hematocrit (HCT), hemoglobin (HB), erythrocytes (RBCs), mean cell volume (MCV), mean corpuscular hemoglobin (MCH), mean corpuscular hemoglobin concentration (MCHC), mean platelet volume (MPV), plateletcrit (PCT), platelet distribution width (PDW), and red blood cell distribution width (RDW). The values set and validated by the automatic instrument through a specific procedure were considered as the reference values for our samples. 

### 2.3. Biochemical Profile

Serum samples were defrosted to room temperature and vortexed for a few seconds, paying attention to avoid the formation of foam. The automated photometer I-Lab Aries Chemical Analyzer (Instrumentation Laboratory S.p.A, Werfen Company, Milano, Italy) was used for the clinical chemistry analysis, according to the manufacturer’s instructions. The analyzed parameters were albumin (ALB), alanine aminotransferase (ALT), alkaline phosphatase (ALP), gamma-glutamyl transferase (GGT), aspartate transaminase (AST), amylase (AMY), cholesterol (CHOL), triglycerides (TRIGs), total bilirubin (BILT), total protein (TP), creatinine (CREA), urea (UREA), iron (FE), magnesium (MG), potassium (K), calcium (CA), sodium (NA), phosphorus (PHOS), and chlorine (CL). The literature provided the reference values for the parameters analyzed [31].

### 2.4. Serum Protein Electrophoresis

For serum proteins’ electrophoresis, a semi-automatic multiparametric biochemical analyzer (HYDRASYS LC SEBIA, HYGRAGEL PROTEIN 15/30 kit and densitometer) was used to separate the protein fractions in an alkaline medium (pH 9.2) on 0.8% agarose gel, stained with a solution by Amidoscharwz. The protidogram was able to differentiate between albumin (ALB), α-Globulins (α-G), β-Globulins (β-G), and γ-Globulins (γ-G) fractions. 

### 2.5. Innate Immunity Analysis

The different innate immunity parameters in the serum samples were quantified by commercial kits (Bioassay Technology Laboratory, Shanghai, China) following the provided instructions: “Bovine Tumor Necrosis Factor Alpha ELISA Kit” (standard curve range: 10–3000 ng/L; sensitivity: 5.56 ng/L), “Bovine Interleukin 6 ELISA Kit” (standard curve range: 20–6000 ng/L; sensitivity: 10.5 ng/L), “Bovine Haptoglobin ELISA Kit” (standard curve range: 3–900 µg/mL; sensitivity: 1.69 µg/mL), and “Bovine Serum Amyloid A ELISA Kit” (standard curve range: 0.4–40 µg/mL; sensitivity: 0.054 µg/mL). Lysozyme titration was determined as previously described [4]. The serum sample was incubated in contact with *Micrococcus lysodeikticus* (ATCC4698), a microorganism particularly sensitive to the lytic activity of lysozyme, incorporated in agar gel. The presence of lysozyme activity was highlighted after incubation at 37 °C in a humid chamber by the appearance of a lysis halo of the germ around the sample well. The concentration of lysozyme, proportional to the diameter of the clarification ring observable around the well, was measured on the basis of a standard curve obtained by the incubation of a known quantity of lysozyme. 

### 2.6. Statistical Analysis

The sample size was defined for an ANOVA study design with repeated measures within factors, considering the following parameters: an alfa error of 0.05, effect size of 0.25, power of 0.95, correlation among repeated measures of 0.5, and 4 measurements (T1, T2, T3, and T4). For each parameter, the data were expressed as mean ± standard deviation. To evaluate the effect of time, a generalized linear model (GLM) procedure with repeated measures was performed using SAS (version 9.4; SAS Institute, Cary, NC, USA). Turkey’s test was used to determine the significance of differences among the mean values at α = 0.05.

## 3. Results

The blood count revealed variations in some parameters at various sampling times. Specifically, significant reductions in the values of white blood cells at T2 and T4, reductions in hemoglobin and platelets at T4, and increases in lymphocytes at T2 were observed. Corpuscular indices (MCH, MCHC, and MPV) behaved in the same way as the values of hemoglobin and platelets (Table 1).

Some relevant alterations concerning the animals’ metabolism were found when observing the analyzed biochemical parameters (Table 2). In particular, ALP, ALT, AMY, AST, CA, CHOL, CL, MG, and TP were the parameters that showed the most significant variations at the end of the study period with respect to their arrival at the genetic center. 

A significant increase (*p* < 0.0001) was identified from T1 to T4 for all the parameters analyzed by serum protein electrophoresis (Table 3). 

## 4. Discussion

Beef calves are generally subjected to various changes in their social and physical environments in the first few months of their lives. Transport, normally occurring in the first weeks of life, is a challenge for young calves as, in this moment, they are highly susceptible to microorganisms, against which they have no colostral antibodies. Weaning involves the loss of social contact with the dam and therefore of suckling behavior. Young animals are also usually subjected to new housing and dietary changes. The introduction into a new physical environment can also interfere with the animals’ ability to know the location of resources (e.g., food and water), and likewise to recognize their group members, which can generate social stress. While these stressors may be presented separately, what usually occurs is an overlap of several or even all of them [24]. 

Weaning and housing are known to increase the total leukocyte number [11,22,26] due to fluctuations in the number of subpopulations of leukocytes, especially neutrophils. At the same time, however, a reduction in lymphocyte subsets is observed during stressful events, most likely attributable to a redistribution of these cells to immune compartments or tissues from the peripheral circulation [26]. On the contrary, we observed a decrease in the number of white blood cells after weaning compared to during the arrival at the genetic center, as well as a decrease in the number of neutrophils, even if not significant. The lymphocyte subpopulation, instead, increased 7 days after their arrival at the center and then decreased and returned to baseline values after weaning. This seems to highlight that, in this study, weaning and the new housing do not elicit a stress response measurable with the investigated parameters in calves. 

The red blood cell number and hematocrit percentage in calves only showed negligible changes, suggesting no negative effects of weaning or housing on such variables. Only the hemoglobin concentration was decreased, but remained within the desired physiological values for calves. 

Regarding the chemical chemistry parameters, although significant alterations in some analytes were observed over the analyzed time, all parameters remained within the reference range of the specie. Abnormal blood electrolyte parameters are generally caused by nutrition problems [32]. In our study, the change in animal feeding compared to the farms of origin could explain the modification of Ca, Cl, and Mg blood concentrations; likewise, the significant decrease in ALP [15]. Conversely, the significant increase in ALT and AST denoted the involvement of liver function. The liver plays a major role in responding to inflammatory conditions with a non-specific general response known as the acute phase response (APR) [33].

Inflammation, infection, and tissue damage activate the innate acute phase response [34]. During the APR, a series of changes in the hematological parameters’ values occur, such as a decrease in blood plasma, low- and high-density lipoprotein-bound cholesterol and leukocyte numbers, increased values of the adrenocorticotrophic hormone and glucocorticoids, complement and blood coagulation system activations, decreased serum levels of iron, zinc, calcium, vitamin A, and α-tocopherol, as well as a modification in the concentration of several plasma proteins, such as acute phase proteins [35], largely due to changes in hepatic metabolism. Indeed, the pattern of protein synthesis by the liver is drastically altered within a few hours after the activation of the response, resulting in an increase in positive acute phase proteins (APPs) [33] and a reduction in the synthesis of normal blood proteins, such as prealbumin, albumin, transferrin, retinol-binding protein, and cortisol-binding globulin, which represent negative APPs.

APPs synthesis is stimulated by the cytokines released from macrophages and other cell types that are transported to the liver where the APPs are synthesized by hepatocytes [34]. The positive APPs include C-reactive protein, serum amyloid A (SAA), and haptoglobin, and they are released by hepatocytes after cytokine stimulation. Blood levels of APP, such as SAA and haptoglobin, increase in the case of animals suffering from inflammatory or infectious diseases, but also in the case of stressed animals [33]. Accordingly, SAA, haptoglobin, and the concentration of other APPs may represent useful stress markers in cattle and other species [36,37]. Previous studies have found that weaning stress as well as transportation and a novel environment are responsible for the activation of the adaptation system. In particular, increased blood concentrations of APP have been demonstrated in calves following exposure to different stressors, which may include abrupt weaning [37], slippery floors [38], and the transportation and mixing of animals [36]. Although the mechanism behind the stress-induced APP response is not known, it may be attributed to hypothalamus—pituitary–adrenal axis activation, causing increased glucocorticoids production.

The APR is believed to be an exclusive inflammation and/or infection biomarker. Inflammatory cytokines, such as IL-1, IL-6, TNF-α, and IFN-γ, are the main mediators of immunological and pathological responses to stress and infection. Our results show a decrease in APTO and SAA and an increase in TNF-α, even if not significant, and in IL-6 after calves weaning. The increase in TNF-α and IL-6 concentrations in calves after their introduction into a new environment and after the influence of different stressors is in agreement with the other studies [6].

Moreover, the acute phase response and, in particular, the effects of IL-1, IL-6, and TNF-α was also associated with the changes in lipid and glucose metabolism, such as cholesterol reduction [39,40], which can explain the significant decrease in cholesterol and the modification of triglycerides observed in our study. Stressful events also resulted in anorexia and energy-store depletion in cattle [27].

Blood components, including serum proteins, are potential health indicators in dairy cows [41]. Pathological and physiological states can result in a modulation of globulin and albumin concentrations in the blood. Thus, the evaluation of their concentrations can be useful for the evaluation of patho-physiological conditions affecting animal welfare and can represent an initial screening method to identify animals that need to be submitted to further clinical investigations. Albumin concentration variations can indicate impaired liver function due, for example, to inflammatory conditions [42], while the concentration of total serum globulin has been reported as an indicator of the animal’s immune response [43]. In this study, the increase in globulins, including gamma-globulin, indicated an immune response of the animals, possibly related to their vaccination, while the increase in albumin supported the hypothesis of liver involvement in an acute phase response.

Lysozyme is an anti-bacterial enzyme that can be found in the body tissues, fluids, and secretions of humans and animals. It plays a crucial role in innate defense mechanisms [44,45] and it is a constituent of primitive non-specific defense mechanisms associated with the monocyte–macrophage system phylogenetically older than the more specific lymphocyte–plasma cell–immunoglobulin system [46]. Sotirov et al. [47] studied serum lysozyme concentrations in different cow breeds, and all the animals investigated, except for four animals, showed values less than 1 µg/mL (dramatically elevated serum lysozyme concentrations are exceptional in cattle). Furthermore, other studies showed that the serum lysozyme concentration is influenced genetically [48,49], decreases with age [49], and is breed-related in cows [49,50]. The results obtained in our study show lysozyme values above 1 µg/mL. This value may be attributed to the young age of the examined animals. The significant increase from T1 to T4 can be justified by the entry of the animals into a new environment and therefore by the different compositions of microbial flora, which required the activation of immune defenses.

## 5. Conclusions

APR is responsible for different effects on a stressed animal, which can include behavioral changes, an increase in body temperature, and the production of pro-inflammatory cytokines and APP. APR is also critical to an animal’s maintenance of homeostasis. Homeostasis is necessary for optimal livestock production, where animal growth as well as reproduction is a priority. Gender, breed, and temperament are known to influence APR. Such variations may also modulate stress response, which underlies a different impact of stress on the innate immune function [5].

Husbandry management practices, including introduction into a novel environment, mixing with other animals, and weaning, represent physical and psychological stressors for calves that are responsible for the activation of APR. In this study, animal behavior was not taken into consideration; however, we only focused on the alterations in blood parameters. The findings show a mild response to adaptation stress by calves, mainly represented by an increase in the values of IL-6 and TNF-α, considered to be two of the main cytokines mediating stress responses. We can therefore hypothesize that the animals included in this study were not in a state of suffering or malaise but activated their adaptation systems in response to the stimuli, such as their introduction into a new environment, the creation of new social settings, weaning, and vaccination. The results of this study are most likely linked to the behavior and temperament of these breeds, as well as to the fact that they are carefully handled and cared for by humans. The animal’s perception affects the levels of stress concerning handling and restraint, while its temperament and reaction impact human–animal interactions. Moreover, positive human interactions can have significant beneficial effects on the animals [51,52,53].

## Figures and Tables

**Figure 1 vetsci-10-00545-f001:**
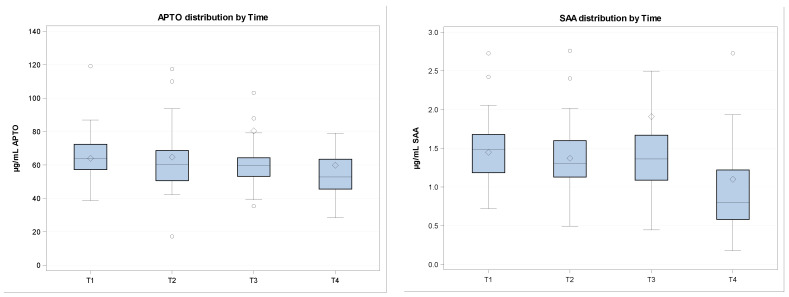
In the figure, cytokines and APP quantifications are shown at different sampling times. Data are normalized. (**a**) APTO (haptoglobin) distribution by time; (**b**) SAA (serum amyloid) distribution by time; (**c**) IL-6 (interleukin-6) distribution by time; (**d**) TNF (tumor necrosis factor α) distribution by time.

**Table 1 vetsci-10-00545-t001:** Means ± SD of blood parameters at different sampling times.

	T1	T2	T3	T4
BA %	0.44 ± 0.26	0.39 ± 0.26	0.45 ± 0.29	0.41 ± 0.26
EO %	0.51 ± 1.12	0.33 ± 0.78	0.19 ± 0.30	0.32 ± 0.27
WBC (m/mm^3^)	11.7 ± 2.5	* 9.77 ± 2.69	10.52 ± 2.60	* 10.13 ± 2.00
RBC (m/mm^3^)	10.2 ± 1.18	9.73 ± 1.24	10.17 ± 1.23	9.92 ± 1.14
HB (g/dL)	12.8 ± 2.01	12 ± 2	12.16 ± 1.46	** 11.69 ± 1.04
HCT %	34.2 ± 6.05	32.8 ± 6.12	33.47 ± 4.98	34.36 ± 3.34
LINF %	45.3 ± 13.2	** 51.3 ± 12.4	46.31 ± 8.04	48.34 ± 8.31
MCH (pg)	12.5 ± 1.07	12.3 ± 0.93	*** 11.92 ± 0.7	*** 11.79 ± 0.95
MCHC (g/dL)	37.6 ± 3.01	36.8 ± 2.77	36.47 ± 2.21	**** 34.03 ± 1.52
MCV (fL)	33.6 ± 3.39	33.6 ± 3.45	32.93 ± 2.44	34.87 ± 2.97
MON %	6.4 ± 3.36	6.85 ± 2.51	6.63 ± 1.57	6.18 ± 1.32
MPV (fL)	9.96 ± 0.77	9.77 ± 0.85	9.94 ± 0.76	** 9.48 ± 0.67
NEU %	47.3 ± 15.2	41.1 ± 13.9	46.39 ± 7.82	44.73 ± 8.17
PCT %	0.59 ± 0.39	0.72 ± 1.16	0.51 ± 0.28	0.33 ± 0.11
PDW	8.79 ± 1.96	8.62 ± 1.89	9.13 ± 1.69	8.86 ± 1.64
PLT (m/mm^3^)	578 ± 347	549 ± 301	505 ± 245	*** 341 ± 106
RDW	17.2 ± 2.11	16.9 ± 2.29	17.24 ± 2.02	16.30 ± 1.13

“BA”: basophil; “EO”: eosinophil; “WBCs”: leukocytes; “RBCs”: erythrocytes; “HB”: hemoglobin; “HCT”: hematocrit; “LINFs”: lymphocytes; “MCH”: mean corpuscular hemoglobin; “MCHC”: mean corpuscular hemoglobin concentration; “MCV”: mean cell volume; “MONs”: monocytes; “MPV”: mean platelet volume; “NEUs”: neutrophils; “PCT”: plateletcrit; “PDW”: platelet distribution width; “PLTs”: platelets; “RDW”: red blood cell distribution width. * *p* < 0.01, significant difference with respect to T1. ** *p* < 0.05, significant difference with respect to T1. *** *p* < 0.001, significant difference with respect to T1. **** *p* < 0.0001, significant difference with respect to T1.

**Table 2 vetsci-10-00545-t002:** Means ± SD of blood clinical chemistry values at different sampling times.

	T1	T2	T3	T4
ALP (U/L)	575 ± 271	**** 246 ± 124	**** 240 ± 103	**** 180 ± 57
ALT (U/L)	17.2 ± 5.45	15.7 ± 5.41	**** 23.9 ± 5.2	**** 45 ± 7.8
AMY	32.9 ± 8.48	34.8 ± 10.1	**** 40.5 ± 12.4	**** 44.2 ± 12.4
AST (U/L)	87 ± 24.5	**** 73 ± 18.3	**** 117 ± 34	**** 111 ± 17
BILT (mg/dL)	0.25 ± 0.13	0.23 ± 0.12	** 0.17 ± 0.09	0.23 ± 0.18
CA (mg/dL)	9.45 ± 3.56	8.06 ± 2.82	7.93 ± 2.3	**** 13.18 ± 6.74
CHOL (mg/dL)	127 ± 49.4	**** 91.1 ± 40.2	**** 101 ± 28.7	**** 65 ± 24.2
CL (mEq/L)	104 ± 9.59	**** 99.3 ± 4.26	107 ± 9	**** 99.2 ± 7.4
CREA (mg/dL)	0.95 ± 0.34	**** 1.12 ± 0.24	0.89 ± 0.25	0.87 ± 0.21
FE (µg/dL)	120 ± 86.9	**** 86.2 ± 57.4	**** 153 ± 66.3	136 ± 39.4
GGT (U/L)	22.7 ± 7.16	19.9 ± 7.32	**** 28.4 ± 12.14	19.7 ± 10.4
K (mg/dL)	8.88 ± 1.91	12.3 ± 17.6	8.15 ± 1.17	7.2 ± 2.3
MG (mg/dL)	1.66 ± 0.67	1.87 ± 1.03	**** 2.85 ± 1.76	**** 2.71 ± 0.96
NA (mg/dL)	147 ± 10.7	**** 140 ± 4.84	**** 152 ± 11.1	146 ± 7.9
PHOS (mg/dL)	7.89 ± 3.15	7.61 ± 3.14	**** 11.24 ± 3.37	8.37 ± 1.88
TP (g/dL)	5.32 ± 0.81	5.04 ± 0.64	**** 5.92 ± 0.72	**** 6.84 ± 0.57
TRIG (mg/dL)	20.1 ± 13.3	* 13.4 ± 6.75	17.6 ± 7.67	16.04 ± 6.94
UREA (mg/dL)	13.2 ± 5.75	*** 19.3 ± 10	13.87 ± 7.06	14.33 ± 6.69

“ALP”: alkaline phosphatase; “ALT”: alanine aminotransferase; “AMY”: amylase; “AST”: aspartate transaminase; “BILT”: total bilirubin; “CA”: calcium; “CHOL”: cholesterol; “CL”: chlorine; “CREA”, creatinine; “FE”: iron; “GGT”: gamma-glutamyl transferase; “K”: potassium; “MG”: magnesium; “NA”: sodium; “PHOS”: phosphorus; “TP”: total protein; “TRIGs”: triglycerides; “UREA”: urea. * *p* < 0.01, significant difference with respect to T1. ** *p* < 0.05, significant difference with respect to T1. *** *p* < 0.001, significant difference with respect to T1. **** *p* < 0.0001, significant difference with respect to T1.

**Table 3 vetsci-10-00545-t003:** Means ± SD of electrophoresis values at different sampling times.

	T1	T2	T3	T4
ALB (g/dL)	2.67 ± 0.51	2.49 ± 0.4	2.87 ± 0.42	* 3.07 ± 0.43
α-G (g/dL)	1.03 ± 0.16	1.03 ± 0.14	* 1.21 ± 0.15	* 1.26 ± 0.16
β-G (g/dL)	0.76 ± 0.2	0.71 ± 0.14	0.69 ± 0.11	* 0.87 ± 0.23
γ-G (g/dL)	0.85 ± 0.26	0.82 ± 0.23	* 1.15 ± 0.30	* 1.63 ± 0.50

“ALB”: albumin; “α-G”: alpha globulin; “β-G”: beta globulin; “γ-G”: gamma globulin. * *p* < 0.0001, significant difference with respect to T1. Quantified acute phase proteins (haptoglobin (APTO) and serum amyloid (SAA)) and TNF-α do not reveal any fluctuations over time, while IL-6 and lysozyme show a significant increase at T4, with respect to the arrival day (Table 4 and Figure 1).

**Table 4 vetsci-10-00545-t004:** APPs, cytokines, and lysozyme quantifications at different sampling times. The results are expressed as mean ± SD.

	T1	T2	T3	T4
APTO (μg/mL)	63.98 ± 13.89	64.73 ± 23.09	80.40 ± 92.62	59.86 ± 37.55
SAA (μg/mL)	1.45 ± 0.42	1.37 ± 0.45	1.91 ± 2.9	1.1 ± 1.4
IL_6 (ng/L)	259 ± 101	273 ± 80	335 ± 220	* 503 ± 368
TNF-α (ng/L)	108 ± 25	111 ± 33	156 ± 169	161 ± 156
LYSOZYME (μg/mL)	1.15 ± 0.58	0.96 ± 0.5	1.19 ± 1.34	* 1.99 ± 1.41

“APTO”: haptoglobin; “SAA”: serum amyloid; “IL_6”: interleukin 6; “TNF-α”: tumor necrosis factor. * *p* < 0.0001, significant difference with respect to T1.

## Data Availability

The data presented in this study are available on request from the corresponding author.

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
