# Peer review of "Is the Introduction into a New Environment Stressful for Young Bulls?"

_vetsci, 2023, doi:10.3390/vetsci10090545_

Round 1

Reviewer 1 Report

It is not clear to me how your stats were conducted. This is not a formal study with treatments. Also, many of the abbreviations are not defined. I suggest going to an English translation service for help with this. Many items were analyzed but not all discussed. All abbreviations need to be defined

please see above

Author Response

Reviewer 1

It is not clear to me how your stats were conducted. This is not a formal study with treatments. Also, many of the abbreviations are not defined. I suggest going to an English translation service for help with this. Many items were analyzed but not all discussed. All abbreviations need to be defined.

Authors’ response: Thank you for your suggestion. We revised the part concerning the stats by adding the part of the data analysis description which was missing for a typ and defining the various abbreviations.

Regarding the discussion of the analyzed items, we focused our attention on those parameters which showed changes during the study period. Concerning the English was revised by a native language.

Reviewer 2 Report

Your paper indicates that your bulls in your genetics center were not highly stressed by being brought into a new environment.  Your results may not apply to bulls in other situations. Both genetics and previous experiences will have an effect on how cattle will react when they are introduced to a new environment. Some breeds of cattle will have a bigger reaction when they are handled and restrained. Cattle that have lots of quiet and positive interactions with people will also e less stressed in a new place.

Line 1 - Change the title to: Is Introduction into a New Environment Stressful for Young Bulls?

Line 127 - Provide a few sentences of explanation on how these bulls are typically raised. Do they have lots of contact with people? This may reduce stress.

Line 166 - It would be really helpful to provide some description of how the bulls behaved after they arrived at the Control Station. Did they remain behaviorally calm or did they pace and vocalize? Agitated behaviors will increase stress levels.

Line 453 - In the conclusions, you hypothesized that the bulls did not have suffering or malaise.It is important to emphasize that this statement may be true for your particular bulls, but it may not be true for bulls in another situation.

The following references may be helpful -

Chase, C.C. et al. (2017) Evolution of tropically adapted straight bred and crossbred beef cattle: Cortisol concentration and measurement of temperament during weaning and transport, J. Animal Sci. 95:5253-5262.

Fordyce, G. et al. (1985) Temperament and bruising in Box indicus cross cattle, Australian Journal of Experimental Agriculture, 25:283-288.

Grandin, T. and Shirley, C. (20105) How do farm animals react and perceive stressful situations such as handling, restraint, and transport? Animals 5(4):1233-1251.

Fine

Author Response

Reviewer 2

Your paper indicates that your bulls in your genetics center were not highly stressed by being brought into a new environment.  Your results may not apply to bulls in other situations. Both genetics and previous experiences will have an effect on how cattle will react when they are introduced to a new environment. Some breeds of cattle will have a bigger reaction when they are handled and restrained. Cattle that have lots of quiet and positive interactions with people will also e less stressed in a new place.

1: Line 1 - Change the title to: Is Introduction into a New Environment Stressful for Young Bulls?

Authors’ response: Thank you for your suggestion, we changed the title as you suggested.

2: Line 127 - Provide a few sentences of explanation on how these bulls are typically raised. Do they have lots of contact with people? This may reduce stress.

Authors’ response: Thank you for your suggestion. We added a part on the text relating to the observation raised.

3: Line 166 - It would be really helpful to provide some description of how the bulls behaved after they arrived at the Control Station. Did they remain behaviorally calm or did they pace and vocalize? Agitated behaviors will increase stress levels.

Authors’ response: Thank you for your interest in the subject and for the inspiration you given us, however animal behavior was not taken into consideration in this study. We focused only on the alterations of blood parameters, without deepening the behavioral pattern of bulls.

4: Line 453 - In the conclusions, you hypothesized that the bulls did not have suffering or malaise.It is important to emphasize that this statement may be true for your particular bulls, but it may not be true for bulls in another situation.

Authors’ response: Thank you for your precisation. We provided for better explanation of this concept in the text of the paper.

5: The following references may be helpful -

Chase, C.C. et al. (2017) Evolution of tropically adapted straight bred and crossbred beef cattle: Cortisol concentration and measurement of temperament during weaning and transport, J. Animal Sci. 95:5253-5262.

Fordyce, G. et al. (1985) Temperament and bruising in Box indicus cross cattle, Australian Journal of Experimental Agriculture, 25:283-288.

Grandin, T. and Shirley, C. (20105) How do farm animals react and perceive stressful situations such as handling, restraint, and transport? Animals 5(4):1233-1251.

Authors’ response: Thank you for your suggestion. We added these paper as references.

Reviewer 3 Report

Dear Authors

The study presented here is a relatively simple examination of markers in the blood/serum of young cattle that are recognised as indicators of stress. The observations made in the study provide an answer to the question posed in the title of the manuscript.

The young cattle involved in the study have been subjected to stress events. Intuitively, one would expect to observe changes in the amounts of these markers following these events.

General Comments.

The introduction to the study report provides sufficient background and information about the study area.

The statistical analysis of the data is appropriate in this case.

The results are clearly described, the use of tables is good and the tables are clearly set out, and the whisker and box plots provide information in a format that is digested easily.

The conclusions the authors have drawn from the findings they have made are sensible. The literature used to support those conclusions is appropriate.

The literature cited by the authors is appropriate.

Specific Comments

Line 19 - insert 'and' before 'evaluation'

Line 42 - re-word the sentence. I suggest - "Stressor" can be defined as any external or internal ... that disrupts body homeostasis.

Line 231 - replace "in" with "at" (to read "at the various ..."

Line 246 should read "... biochemical parameters ..."

Line 264 and 265 - It appears to me that  p<0.05 and P<0.01 should be swapped.

Line 266 and 267 - i don't believe the legend needs the words "indicates a"

Line 442 - the text at the end of this line should be "... as well as ..." 

Line 452 - remove "considered ones ..." and replace with "considered to be two ..."

Line 455 - the word "some" can be removed

I suggest these suggestions be considered carefully as they are aimed at improving the quality of the manuscript.

Dear Authors

The study presented here is a relatively simple examination of markers in the blood/serum of young cattle that are recognised as indicators of stress. The observations made in the study provide an answer to the question posed in the title of the manuscript.

The young cattle involved in the study have been subjected to stress events. Intuitively, one would expect to observe changes in the amounts of these markers following these events.

General Comments.

The introduction to the study report provides sufficient background and information about the study area.

The statistical analysis of the data is appropriate in this case.

The results are clearly described, the use of tables is good and the tables are clearly set out, and the whisker and box plots provide information in a format that is digested easily.

The conclusions the authors have drawn from the findings they have made are sensible. The literature used to support those conclusions is appropriate.

The literature cited by the authors is appropriate.

Specific Comments

Line 19 - insert 'and' before 'evaluation'

Line 42 - re-word the sentence. I suggest - "Stressor" can be defined as any external or internal ... that disrupts body homeostasis.

Line 231 - replace "in" with "at" (to read "at the various ..."

Line 246 should read "... biochemical parameters ..."

Line 264 and 265 - It appears to me that  p<0.05 and P<0.01 should be swapped.

Line 266 and 267 - i don't believe the legend needs the words "indicates a"

Line 442 - the text at the end of this line should be "... as well as ..." 

Line 452 - remove "considered ones ..." and replace with "considered to be two ..."

Line 455 - the word "some" can be removed

I suggest these suggestions be considered carefully as they are aimed at improving the quality of the manuscript.

Author Response

Reviewer 3

Dear Authors

The study presented here is a relatively simple examination of markers in the blood/serum of young cattle that are recognised as indicators of stress. The observations made in the study provide an answer to the question posed in the title of the manuscript.

The young cattle involved in the study have been subjected to stress events. Intuitively, one would expect to observe changes in the amounts of these markers following these events.

General Comments.

The introduction to the study report provides sufficient background and information about the study area.

The statistical analysis of the data is appropriate in this case.

The results are clearly described, the use of tables is good and the tables are clearly set out, and the whisker and box plots provide information in a format that is digested easily.

The conclusions the authors have drawn from the findings they have made are sensible. The literature used to support those conclusions is appropriate.

The literature cited by the authors is appropriate.

Specific Comments

Line 19 - insert 'and' before 'evaluation'

Authors’ response: Thank you for your suggestion. We have changed the text

Line 42 - re-word the sentence. I suggest - "Stressor" can be defined as any external or internal ... that disrupts body homeostasis.

Authors’ response: Thank you for your suggestion. We have changed the text

Line 231 - replace "in" with "at" (to read "at the various ..."

Authors’ response: Thank you for your suggestion. We have changed the text

Line 246 should read "... biochemical parameters ..."

Authors’ response: Thank you for your suggestion. We have changed the text

Line 264 and 265 - It appears to me that p<0.05 and P<0.01 should be swapped.

Authors’ response: Thank you for your suggestion. We have changed the text

Line 266 and 267 - I don't believe the legend needs the words "indicates a"

Authors’ response: Thank you for your suggestion

Line 442 - the text at the end of this line should be "... as well as ..."

Authors’ response: Thank you for your suggestion. We have changed the text

Line 452 - remove "considered ones ..." and replace with "considered to be two ..."

Authors’ response: Thank you for your suggestion. We have changed the text

Line 455 - the word "some" can be removed

Authors’ response: Thank you for your suggestion. We have changed the text

I suggest these suggestions be considered carefully as they are aimed at improving the quality of the manuscript.

Round 2

Reviewer 1 Report

Thank you for addressing my edits and concerns.

I have a suggestion- define the variables using superscripts and put them at the bottom of the tables where your P values are located.

Author Response

Thank you for the suggestion; we proceeded to modify the text as you suggested.

Reviewer 2 Report

Accept Minor Revision

Line 37 - Add after the word calves, "raised in close association with people."

Fine

Author Response

(The authors gave the same response as above.)
